# Gamification as an Educational Approach for Oncological Patients: A Systematic Scoping Review

**DOI:** 10.3390/healthcare11243116

**Published:** 2023-12-07

**Authors:** Andrea Poliani, Silvia Gnecchi, Giulia Villa, Debora Rosa, Duilio F. Manara

**Affiliations:** 1Center for Nursing Research and Innovation, Faculty of Medicine and Surgery, Vita-Salute San Raffaele University, 20132 Milan, Italy; poliani.andrea@hsr.it (A.P.); rosa.debora@unisr.it (D.R.); manara.duilio@hsr.it (D.F.M.); 2IRCSS San Raffaele Scientific Institute, 20132 Milan, Italy; gnecchi.silvia@hsr.it

**Keywords:** gamification, education, cancer patient, cancer survivor, oncology

## Abstract

Background: Education plays a pivotal role in the care of oncological patients, reducing health costs, hospital readmission, and disease relapses. Education can be supportive in achieving multiple outcomes, improving symptom control and quality of life. A new approach is emerging in patient education: gamification. Gamification was defined as the “use of game elements in non-game contexts”, including the application of games in serious contexts. The aim of this review is to explore the use of gamification in the oncology setting. Methods: A systematic scoping review was conducted in the MEDLINE, CINAHL, PsychINFO, Embase, Scopus, and Cochrane Library databases using the JBI guidelines. Results: The 13 included reports were critically appraised by two reviewers independently. It seems that gamification could be effective both in prevention and cancer treatments. Gamification also seems to improve chemotherapy-induced nausea and vomiting management, quality of life, and reduced anxiety levels in different cancer groups. Moreover, gamification seems effective in improving self-care in cancer patients, regardless of gender, age, and ethnicity. Conclusions: Gamification improves patient engagement and biopsychosocial outcomes and could represent a valid approach to cancer patient education; however, it is not a substitute for healthcare professionals, who remain the leaders in the education process.

## 1. Introduction

The cancer burden continues to grow worldwide, causing serious damage to the physical, psychological, and social states of individuals and their caregivers. Although cancer still represents the second leading cause of death, five-year survival rates have increased globally from 49 to 68% in the last ten years [1]. The global cancer load is expected to be 28.4 million cases in 2040, a 47% rise from 2020 [2]. Nowadays, in Europe, the estimated numbers of new cancer diagnoses are approximately 1.4 million in males and 1.2 million in females, with over 710,000 estimated cancer deaths in males and 560,000 in females [3]. 

Since cancer involves problems from a cognitive, psychological, and functional point of view, education of patients and caregivers could enhance both clinical and biopsychosocial outcomes [4]. Patient education, a branch of health education, is defined as the process of enabling individuals to make informed decisions. Healthcare professionals are responsible for encouraging adherence to treatments and promoting correct lifestyles [5]. Through patient education, behavioral change is promoted, which consists of a complex process requiring more than the simple acquisition of knowledge [6]. In particular, nurses seem to represent the principal stakeholders in patient education, increasing convalescents’ knowledge, skills, and awareness of health and health status [7]. For these reasons, empowering patient education is gaining ground as a promising approach to improving outcomes and reducing complications [8,9]. 

Among empowering strategies, gamification seems to represent a new supporting approach in patient education [10,11]. Gamification is defined as the “use of game elements in non-game contexts” [12]. Though simple, this definition clarifies the difference between typical games and serious games. The first use has the aim of fun entertainment only, while the second use is designed to promote learning, motivate action, and change users’ attitudes and behaviors [13]. Gamification, to be defined as such, should include several game elements, for example, self-representation with avatars, levels, rewards, competition, parallel communication systems, and time pressure [12]. In healthcare systems, serious games serve several goals, ranging from training healthcare providers to the promotion of healthy lifestyles in the broader patient population [10]. In the caring field, gamification is often used as a technological approach supporting the patient education process. Gamification and serious games have already been applied in the noncommunicable diseases field, such as diabetes [14] and cardiovascular disorders [15].

The necessity to conduct a systematic scoping review comes from the rising numbers of cancer and the need, as healthcare providers, to stay up to date on the latest care approaches in the oncological field. The aim of this review is to map, explore, and examine if gamification represents an effective approach to educating cancer patients or cancer survivors. In this way, it will be possible to identify gaps in evidence-based practices to enhance cancer patient education.

## 2. Materials and Methods

Systematic scoping reviews are designed to map key concepts and examine studies in a research area to provide an overview of the extent and nature of the current literature. This review should be considered a systematic scoping review due to its rigorous methodology and the double-blind quality appraisal of selected sources.

This review aims to provide an overview of research on gamification in oncology education. 

This systematic scoping review was conducted in line with the Joanna Briggs Institute (JBI) Manual for Evidence Synthesis guidelines [16]. Before writing, a planned search on PROSPERO and JBI Review Register was made to ensure that no review protocols on this topic are in progress. As mentioned in the “JBI Manual for Evidence Synthesis”, this systematic scoping review followed the “PRISMA ScR Checklist” for its construction [17]. Their international relevance and the need to ensure both methodological rigor and transparency in the conduct of the review led to the decision to use the guidelines and, subsequently, the checklist.

In order to examine the extent and nature of gamification as an education approach in the oncological field, the following research question was identified: “To what extent is gamification effective in improving the educational outcomes of cancer patients and survivors in a variety of settings, such as the hospital, the home and the community?”

The research question was structured following the Participant-Problem/Concept/Context (PCC) framework, as suggested in the “JBI Manual for Evidence Synthesis” guidelines [16], to guide and direct the development of the specific eligibility criteria, keywords, and search strategies (Table 1). 

### 2.1. Eligibility Criteria

#### 2.1.1. Inclusion Criteria

Study selection was based on studies that answer the research question and the PCC framework (Table 1) [16]; studies that mention and discuss gamification and/or serious games for educating oncological patients and survivors; and the setting of the studies is the hospital, homes, and the community. This review considered both primary sources of evidence and reviews.

#### 2.1.2. Exclusion Criteria

Exclusion criteria were articles on the creation and development of a serious game without testing in a real population, as they may not provide insights into the effectiveness of gamification on the target population; articles on the creation and development of a serious game without testing in a real population, as they may not provide insights into the effectiveness of gamification on the target population; virtual reality (VR) without gamification, training, and teaching in nursing or medical school settings because they pertain to a context that differs from the chosen one; studies that talk about gaming with an only playful or fun aim without a patient education purpose; and documents that belong to grey literature due to their lack of peer review and challenges in quality assessment.

#### 2.1.3. Limits

During the database searches, some filters were applied: a language filter was applied to English and Italian, and a no-time-restriction filter was used to include all the literature about gamification and oncology. This choice was due to the novelty of the topic.

### 2.2. Search Strategies

A three-step search strategy was conducted as suggested using the JBI guidelines [16]. The first step was an initial limited search on MEDLINE via PubMed and Cumulative Index to Nursing and Allied Health Literature (CINAHL) to analyze the text words and the index terms used to describe the retrieved articles. A second search using all identified keywords and index terms was undertaken across all included databases to identify the pertinent and relevant papers. Thirdly, the reference lists of the selected sources were employed for additional searches. 

Search strategies were developed with the assistance of a university research expert and resulted in the following search categories: Medical Subject Headings (MeSHs), text words, and word variants concerning gamification as well as terms related to oncological patients (Appendix A).

An electronic search was conducted from the databases on November 21/22, 2022 (dates of databases searching) in the following databases: MEDLINE via PubMed, CINAHL, PsycINFO, Excerpta Medica dataBASE (Embase), Scopus, and The Cochrane Library. The search relied upon the hospital and university’s library and services to interrogate the nursing and biomedical electronic databases. All the documents that were not available were requested by the library service if not available in full-text form in the electronic databases. If the library service did not retrieve the document, the reference authors were contacted.

### 2.3. Document Selection

The selection process followed the PRISMA Flow Diagram 2020 [18]. Study selection was divided into two stages, managed with the RAYYAN web app [19]. In the first step, titles and abstracts were screened by two independent reviewers (A.P. and S.G.) for assessment against the inclusion criteria for the review. In the second stage, the full texts of selected citations were assessed in detail against the inclusion criteria and then were read following a peer review process by two independent reviewers (A.P. and S.G.). Any divergence was resolved through discussion or with a third senior reviewer (G.V.). During the selection process, no particular challenges or ambiguities were encountered. All the data were systematically collected, sorted, and deduplicated using the reference manager Zotero. Manual sorting followed.

### 2.4. Methodological Quality

Methodological quality was assessed by two independent reviewers (A.P. and S.G.) using the “Dixon-Woods Prompts for assessing quality in primary research” tool [20]. The decision to use this quality assessment tool was made because it was expected that different study designs would be encountered, including articles related to the development and validation of serious games, some of which may not have validated quality assessment tools. Finally, this tool allowed us to compare the quality ratings of different study designs. Any reviewer disagreements were resolved through discussion or with a third senior reviewer (G.V.).

In this systematic scoping review, all the studies included were analyzed for their methodological quality to make observations about gamification effectiveness in different types of design studies. 

### 2.5. Data Extraction

Metadata were extracted using Zotero software (version 6.0.30) and manually checked upon import. Data were extracted from papers included using a standardized data extraction tool according to the JBI “Manual for Evidence Synthesis” [16]. Due to the specific topic, the data extraction table was also enriched with game elements, oncological settings, key findings, and study limits (Appendix A).

## 3. Results

Initially, 289 documents were retrieved. The entire selection process followed the PRISMA Flow Diagram 2020 [18], which finally resulted in the selection of 13 full-text articles (Figure 1).

Most of the selected articles are primary studies except for an extensive narrative review. The selection consisted of three randomized controlled trials [21,22,23]; one randomized experimental study [24]; two qualitative studies [25,26]; three pilot studies [27,28,29]; one evaluation study [30]; one community advisory board study [31]; and one study with a mixed-method approach [32]. The publication years of the included studies were from 2016 to 2022. Most of the studies were from the USA (n = 7) [22,23,24,28,29,31,32]. The remaining studies were conducted in South Korea (n = 2) [21,27], Australia (n = 1) [25], Greece (n = 1) [33], Canada (n = 1) [26], and the UK (n = 1) [30]. All papers were written in English.

For the content analysis, the 13 included articles were grouped into prevention, educational outcomes, and gamification development categories.

### 3.1. Quality Appraisal

According to the different study designs appraised with the “Dixon-Woods Prompts for assessing quality in primary research” tool [20], most of the included articles are of good quality (>60% as stated by the authors). Quantitative studies varied in methodological quality ranging from 20 to 100%, while qualitative studies all showed a percentage of 80. Two articles showed a weak methodological quality with 20% of positive answers [30,31]. Just one study presented a medium quality of 60% [33] (Table 2). 

### 3.2. Educational Outcomes 

In all the analyzed articles, gamification is considered as an educational approach. Gamification in oncology often has an educational perspective and answers to patients’ needs [21,22,29,32,33]. The principal areas of gamified education are feedback and motivation [23,24,26,27,29], prevention [22,24,25,28], psychological support [21,27], symptom management [31], quality of life [21,32], and self-care [29,33].

#### 3.2.1. Feedback and Behavioral Change

Carcioppolo et al. [24] provided two kinds of feedback: dermatological and dermatological plus motivational groups. In the first intervention group, the participants were given feedback according to their categorization of nevi skills. In the second one, they did not just learn if they were correct or incorrect in melanoma selection, but they also received motivational messages. Feedback facilitated and motivated the learning process, increasing self-efficacy level and motivation in both intervention groups compared with the control (*p* = 0.002). In “Hit the Cancer” by Kim et al. [27], feedback, mood encouragement, and active stimulation were employed as effective educational strategies for breast cancer patients. In the serious game called “Strong Together”, a feedback screen with ideal solutions reinforces advanced cancer patients’ learning [29]. Also, feedback and nurse-led positive reinforcement during serious games were applied instead of complex written materials in older adults with cancer in chemotherapy-induced nausea and vomiting (CINV) prevention. Owing to educational enhancement, the game group reported more preventive behaviors than the control group [23]. Biofeedback emerged to be effective as an educational approach in swallowing therapy sessions in patients with head and neck cancer. During semi-structured interviews, participants who received immediate and simple feedback declared feeling more motivated and engaged [26].

Other educational approaches that emerged from the included studies are behavior change and simulation. In several serious games, the oncological patient simulated, through avatars, the correct behavior in different settings, situations, and health states [21,30]. Behavior modification and decision making were improved through needs and priority identification in patients with cancer [22,29,32]. Facing a problem of health, patients were asked to act in the first person, making decisions to increase and improve wellbeing in the gamified setting [33].

#### 3.2.2. Prevention

Of the thirteen studies included in this review, four assessed the role of gamification in cancer prevention [22,24,25,28]. 

Most of the studies on prevention were conducted in the dermatological field. Maganty et al. [28] developed an online series of games called “Tapamole” for melanoma recognition. This three-arm randomized controlled trial (RCT) included patients with a history of skin cancer or malignant melanoma who were allocated into game, pamphlet, and no-intervention groups. The game group played “Tapamole”, which consists of recognizing a melanoma cancer from different kinds of images shown. The results showed that gamification was as effective as the written pamphlet (specificity *p* = 0.05). This study also showed how gamification can be applied to patients as a valid tertiary prevention intervention in skin cancer. The authors underlined the concept of gamification as an approach to, and an integration of, the educational process. 

Horsham et al. [25] developed a virtual reality game about skin cancer prevention using a feedback and messaging format. The game suggested how and when people should apply sun protection cream. The participants preferred the VR game over other presentation formats, such as videos or brochures, referring to a better connection to the environment in the three-dimensional immersion than in the other educational tools. 

Carcioppolo et al. [24] conducted a randomized experiment about the prevention and early recognition of cutaneous melanoma using different systems (Asymmetry Border Color Diameter Funny looking—ABCDF, Asymmetry Border Color Diameter—ABCD, and Ugly Duckling Sign—UDS) compared with a control group that did not receive any information. The game was made up of twelve stimuli conditions. All conditions required the participant to identify malignant melanomas. The game seemed to increase the participants’ self-efficacy to perform self-screenings and encouraged cancer prevention attitudes and intentions. The game proved the efficacy of the ABCD and UDS systems (*p* < 0.001 and *p* = 0.05, respectively) in identifying melanoma compared with the control group.

Khalil et al. [22] conducted a three-arm single-blinded RCT including a “Re-Mission” serious game addressed to different oncological diseases. This video game revolves around Roxxi, a nanobot injected into the body of virtual cancer patients to help them fight cancer. The experimental groups played the “Re-Mission” video game at high or low challenge levels. In the high-challenge experimental group, there was an increased perception of cancer severity (*p* = 0.005), the low-challenge group exhibited no significant changes, and no differences were observed in the control group. The high-challenge group showed an increased perception of susceptibility to cancer from baseline to the post-test assessment (*p* = 0.002). The role of “Re-Mission” led to an increase in perceived cancer severity and susceptibility in healthy young adults. The use of gamification and, in particular, the “Re-Mission” game, allowed young adults to discover new information about cancer and its prevention, learning more about self-protection.

#### 3.2.3. Psychological Distress

In two studies including women with breast cancer [21,27], anxiety was intended to be defeated through serious games.

The pre/post-test study conducted by Kim et al. [27] recruited women with breast cancer and major depressive disorder (MDD). Anxiety was evaluated with the Beck Depression Inventory (BDI), Beck Anxiety Inventory, and Stress Response Inventory (SRI). In the intervention group, women had to use the “Hit the Cancer” serious game for at least 30 min/day, five days/week; meanwhile, the control group did not play the game. In the post-test assessment, the BDI and SRI scores greatly decreased in the intervention group (*p* < 0.05), demonstrating the role of gamification in reducing anxiety and anxiety symptoms. Gamification could be used to improve depressive symptoms and stress levels in breast cancer patients with depression. Also, Kim et al. [21] analyzed the role of gamification in reducing anxiety levels in women with metastatic breast cancer. Participants were randomly assigned to a mobile serious game group or to a conventional education group 1:1. Both groups were screened with the BDI and the Spielberger State–Trait Anxiety Scale. The intervention group had to play the “ILOVEBREAST” serious game for at least 30 min/day, three times a week. The two groups had no significant changes in the BDI score for anxiety. The authors reported that the time spent on game playing in the intervention group was higher than in the traditional education group (*p* < 0.001) and also reported that the game was found to be difficult to use by half of the sample (56%). 

#### 3.2.4. Chemotherapy-Induced Nausea and Vomiting (CINV) Management

Loerzel et al. [23] structured a randomized controlled trial to examine the “eSSET CINV” serious game to educate patients to reduce and control CINV at home. Participants were all old adults suffering from different cancers who seemed to increase CINV preventive self-management behaviors through the gamified intervention.

#### 3.2.5. Quality of Life

Using gamification in the oncological field could help people suffering from cancer improve their quality of life [21,32]. Reichlin et al. [32] recruited men who had already completed the treatment for localized prostate cancer. All the participants played the “Time After Time” serious game three times and participated in focus group meetings. Participants verified that the game meets the goals of increasing focus on health-related quality of life and providing a new educational avenue to augment patients’ participation in choosing a treatment for prostate cancer. Kim et al. [21] reported that gamification could improve the quality of life (*p* = 0.01) of women with breast cancer who played the “ILOVEBREAST” game.

#### 3.2.6. Self-Advocacy and Engagement

Self-advocacy and engagement represent a crucial topic in many of the included studies. For example, Kondylakis et al. [33] created a serious game called “MyHealthAvatar”. All participants were divided into two groups: one tested the serious game, and the control was treated as usual. The two groups were assessed with the Mini-Mental Adjustment to Cancer and Patient Health Engagement (PHE) Scale to examine dimensions like fighting spirit, anxiety, and avoidance. In particular, subscales “fighting spirit”, “anxious preoccupation”, and “avoidance” were reduced in the experimental group, while “fatalism” was reduced in both groups. The PHE scale detected an improving level of engagement in both groups. Thomas et al. [29] tested the “Strong Together” serious game in which participants decide the character’s behavior and select possible actions that may be difficult or impractical in the player’s reality. The authors recruited women with different types of cancers. The serious game augmented their self-advocacy, with more informed decision making (4.9/6), effective communication (5.0/6), and more connected strength (4.5/6); moreover, it augmented their quality of life based on physical (4.0/5), social (4.6/5), emotional (3.7/5), and functional (3.8/5) factors (higher scores correspond to a higher quality of life or self-advocacy). 

### 3.3. Gamification Development in Oncology

As emerged from the included articles, gamification development in oncology consists of projecting, testing, and implementing health technologies with an educational purpose through a multidisciplinary team [25].

#### 3.3.1. Figures Involved in Gamification

The development of gamification is a multidimensional approach that involves different figures, as reported in the included papers. In fact, some specialists are responsible for patient education and administer serious games, such as psychiatrists [27], oncology nurses, research nurses or general researchers [22,23,25,26,32,33], oncologists, clinical psychologists [29], speech therapists and physical therapists [26], and healthcare professionals in general [21,28,31,33]. On the other hand, for technical development, other specialists are crucial, such as game designers, software developers, social scientists, and computer scientists [24,25,28,30,33] (Table 3).

#### 3.3.2. Game Elements

Gamification presents various game elements often connected to its educational purpose. The main game elements are avatars [21,23,24,25,28,29,30,31,33], levels, rewards and story progression [23,24,25,27,30,31], realistic dialogue and response options [26,29,30], card matching [24,28,32], shooting [22,25,27,33], time pressure, and competition [25,27]. Avatars could allow patients to identify themselves with the game character, therefore making more informed decisions and increasing their engagement (Table 4).

#### 3.3.3. Setting and Duration

Serious games are intended to be played by patients in different settings according to the stage of illness, the kind of treatment undergone, and the patient’s characteristics [33]. In the analyzed articles, all the gamified interventions for oncological patients are digitized and can be applied everywhere [21].

The scenario of the game represented the virtual environment and the graphic interface. It played a determinant role in promoting the educational purpose [21]. Some serious games presented a scenario that reproduced everyday life settings [21,23,30,31]. The serious games were available by tapping a web link [24,31] or by downloading an app on a personal mobile phone [21,27,28,29,30,33]. Only two games were specified to be played at home as educational reinforcement after the hospital discharge [26,31]. Culturally appropriate interventions, content, and settings of serious games are provided for targeted populations such as African–Caribbean men [30] or Hispanic/Latino populations [24]. The mean duration of the gameplay is a few minutes (from 1 to 15 min). In three studies, the game-based learning lasted 30–35 min [21,22,27].

## 4. Discussion

This scoping review aimed to examine the extent and nature of gamification as an educational strategy in oncological settings. It also showed gamification as a new and effective educational approach to prevent, engage, empower, and reduce symptomatology in cancer patients. To the best of our knowledge, this is the first scoping review describing the role of gamification in educating cancer patients. The strength of this review is its stringent exclusion criteria and the quality assessment approach.

### 4.1. Heterogeneous Cancer Diseases

Heterogeneous types of cancer are considered in this scoping review. This element guarantees a complete overview of gamification applications in different oncological fields, including breast [21,27], prostate [30,32], dermatological [24,25,28], head and neck [26], and general cancers [22,29,33]. The authors’ choice to focus on breast, skin, and prostate cancer could be associated with the high incidence of these cancers and the low survival rates [1,34]. Two articles [23,31] were about CINV, which is not a cancer disease, but a complication tightly connected to oncology. 

The patient education process is not simply limited to the period of first diagnosis and the period of end-stage conditions. In fact, too many new elements at once could be overwhelming for cancer patients [23]. In such cases, patients may be not sufficiently engaged in playing serious games effectively because they are often in the “blackout” condition as reported by using the PHE scale [35]. During this condition, patients may be unable to cope with their health condition and are thus less able to contribute to their empowerment and education. 

From the included articles, it emerged that game-based learning was specific according to the disease and the educational approach was tailored according to the kind of cancer. Therefore, gamified interventions are not generalizable to all cancer patients.

### 4.2. Development and Application of Serious Games 

Several of the included articles focus on game development and applicability; on the other hand, some of the articles discuss game effectiveness. In fact, some studies included in their purpose the key element “To develop” [24,25,26,29,30,31].

The effectiveness of existing games in cancer patient education emerged in the “Tapamole” [24,28], “Time after Time” [32], “ILOVEBREAST” [21], “Hit the cancer” [27], “Re-MIssion” [22], and “IManageCancer” [33] games. These serious games have already been developed and, as demonstrated in the papers, their purpose is to evaluate the feasibility, appropriateness, and improvement in outcomes. 

Serious games are organized in different settings inside and outside of the hospital environment. Most games have the potential to reach a wide population owing to digital and online-based educational interventions. The everywhere setting guarantees the possibility to extend and easily spread serious games [29,33].

### 4.3. Technological Gamification

Nowadays, everyone owns at least one digital device with the capability to download a web-based game. In addition, multiplayer, social network, and platform-based features make gamified education enjoyable, immersive, and interactive [21]. Two articles treated serious games in geriatrics [23,31]. While gamified intervention may seem difficult for older adults because of the technological complexity, Koivisto and Malik, in their systematic review conducted in 2021 [36], affirmed the effectiveness of gamification in this specific population [36]. In just one case, the collateral effects of digital serious games were reported on and connected to the prolonged use of VR, such as dizziness and impaired balance [25].

### 4.4. Gamification in Satisfying Oncological Patients’ Needs 

Patients’ needs and educational necessities are widely treated in the included articles [23,26,27,29,30,31,32]. Most studies were focused on early symptom recognition and management. Some authors aimed to reach different nursing-sensitive outcomes (NSOs) [37] and patient-reported outcomes (PROs) [38] such as CINV, dysphagia, the activity of daily living, anxiety, depression, and self-care [23,26,27,29,30,31,32]. These elements are crucial to better involve patients in their clinical pathways and can help healthcare professionals guarantee the best standard practice.

### 4.5. Areas for Improvement and Recommendations

This review examined gamification as an educational approach for oncology patients and survivors. Several areas for improvement and further research emerged.

Game elements in the included studies are often not specific according to cancer types, stages, and patient demographics. Tailored solutions may meet the educational needs of different patient populations. For example, it emerged that the geriatric population could benefit from tailored gamified intervention. It is important to take into account potential challenges related to technology use, cognitive impairment, or sensory deficits [36]. Further research is required to explore the feasibility and acceptability of gamified interventions among elderly oncology patients. Additional research is required to involve patients in a holistic way by enabling patients to provide feedback on the fulfillment of individual needs, such as mental health support, rehabilitation exercises, and post-treatment care. Finally, it is essential to consider the potential negative effects of online serious games, such as dizziness and balance disorders. Therefore, game developers should incorporate safety features to limit prolonged use and provide regular breaks.

## 5. Limitations

This literature search showed evidence of the existing gaps in using gamification as an educational approach in oncological settings.

### 5.1. General Limitations

A limitation of this review is the exclusion of articles published in languages other than Italian or English. This restriction may lead to the omission of valuable research from non-English- or non-Italian-speaking regions, potentially limiting the comprehensiveness of the review.

In some papers, the sample was too specific, including women with an MDD diagnosis or patients with prostate cancer belonging to a specific ethnic group: African–Caribbean men living in the UK [27,30] and Hispanic/Latino populations [24]. Another point of interest is the comparatively high amount of literature from the USA, with most papers on Western cultures. The knowledge gained from these papers may not be transferable to other countries and cultures due to geographical and cultural differences.

In addition, different tools and evaluation scales were employed to measure the serious games’ effectiveness and outcome improvement. Heterogeneous measures made it difficult to compare studies’ results.

### 5.2. Quality Appraisal Limitations

Most of the included articles were determined to be of good quality during the quality appraisal even if the tool used was not specific to each study design. The studies’ design structures were very different, so a non-design-specific tool for the quality appraisal work was employed: “Dixon-Woods Prompts for assessing quality in primary research” tool [20]. Choosing this tool may have hindered a deep and detailed quality appraisal although it enabled the comparison of various study types.

### 5.3. Generalizability of Results

All the papers showed a small population sample except for the study conducted by Carcioppolo et al. [24] with 1205 participants. Due to the different sample sizes, we could not generalize the results and data to the population. Moreover, the data reported by the individual studies did not allow for statistical considerations and analysis.

## 6. Conclusions

In this systematic scoping review, gamification emerged as an innovative approach for oncological patients’ education. In particular, serious games seem to be effective in symptom management, cancer prevention, and quality of life improvement. The use of gamified intervention promoted a reduction in anxiety, depression, and psychological distress, which represent the main NSOs and PROs in the oncological field. Gamification could represent a helpful educational strategy for healthcare professionals to empower patients to make informed decisions. 

### 6.1. Implications and Recommendations for Future Research

The theme of gamification in the oncological field has not been sufficiently investigated. Hence, there is a need to conduct more empirical research. To establish and extend the effectiveness of gamification in the oncology population, a combination of qualitative and quantitative evidence is essential. Larger sample sizes, participants from diverse backgrounds, and longer observation periods are needed for a more comprehensive assessment of whether the advantages of gamified interventions, such as reduced anxiety and improved symptom management, endure over time. Prospective cohort studies and randomized controlled trials should be undertaken to assess the long-term advantages and possible drawbacks of this new education approach. In addition, more socioeconomic evaluations should be performed to estimate the costs of using gamified interventions. Gamification, although it cannot be used as a standalone approach, should be integrated into the educational process and can be employed in conjunction with other traditional educational tools to provide a more comprehensive and effective educational strategy. However, gamification still represents a supporting approach that is not a substitute for healthcare professionals, who should remain the leaders of the whole educational process.

Further research is also needed to explore the training of healthcare professionals in this innovative educational approach, which encompasses understanding mechanics, monitoring patient progress, and adapting educational strategies based on game and patient feedback.

Finally, this scoping review primarily centered on the oncological population. Nevertheless, it is worth noting that cancer patients often receive support from both formal and informal caregivers who could be potentially engaged in the creation of serious games and studies aimed at validating their efficacy.

### 6.2. Final Thought

Gamification in oncology education promises to be a supplementary tool for enhancing patient outcomes. While initial findings are encouraging, a deep and multifaceted research approach is necessary to fully grasp the extent of its influence. Healthcare professionals continue to play a central role in the educational process, and with the appropriate tools and training, gamified interventions can augment their endeavors, resulting in improved patient empowerment and engagement, and better care outcomes.

## Figures and Tables

**Figure 1 healthcare-11-03116-f001:**
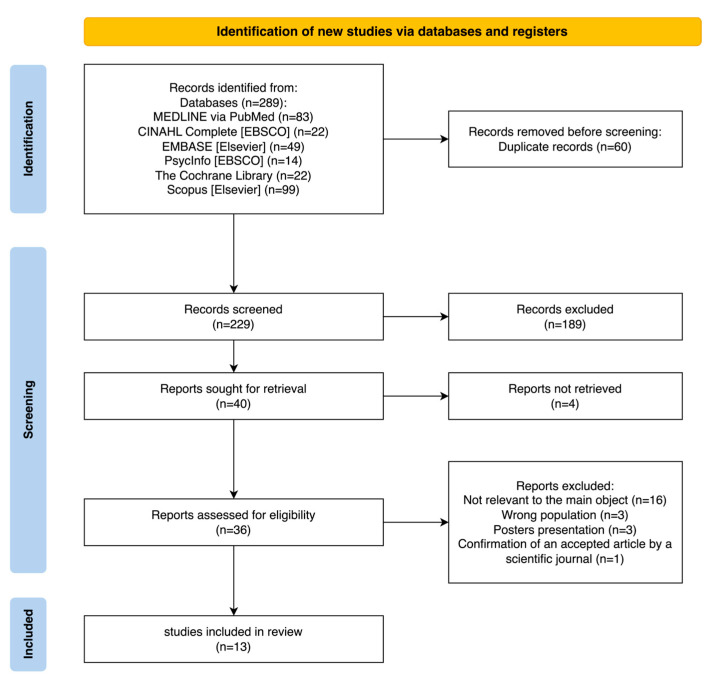
PRISMA Flow Diagram 2020 [18].

**Table 1 healthcare-11-03116-t001:** PCC Framework.

PCC	
Population	Oncological patients with a current diagnosis of tumour and cancer survivors (≥18 years)
Concept	Gamification as defined by Deterding et al. [12]
Context	Hospital, homes and community including cultural factors, geographic locations or gender-based interests

**Table 2 healthcare-11-03116-t002:** Quality Appraisal [20].

Author(s), Year	Are the Aims and Objectives of the Research Clearly Stated	Is the Research Design Clearly Specified and Appropriate for the Aims and Objectives of the Research?	Do the Researchers Provide a Clear Account of the Process by the which Their Findings Were Reproduced?	Do the Researchers Display Enough Data to Support Their Interpretations and Conclusions?	Is the Method of Analysis Appropriate and Adequately Explicated?	YES
Kim, Kim, Shin et al. 2018 [21]	Y	Y	Y	Y	Y	100%
Kim, kim, Hwang et al. 2018 [27]	Y	Y	Y	Y	Y	100%
Carcioppolo et al. 2022 [24]	Y	Y	Y	Y	Y	100%
Horsham et al. 2021 [25]	Y	Y	Y	Y	N/A	80%
Maganty et al. 2018 [28]	Y	Y	Y	Y	Y	100%
Khalil et al. 2016 [22]	Y	Y	Y	Y	Y	100%
Loerzel, Clochesy and Geddie, 2018 [31]	Y	N	N/A	N/A	N	20%
Loerzel, Clochesy and Geddie, 2020 [23]	Y	Y	Y	Y	Y	100%
Reichlich et al. 2011 [32]	Y	Y	Y	Y	Y	100%
Cosma et al. 2016 [30]	Y	N	N/A	N/A	N/A	20%
Thomas et al. 2019 [29]	Y	Y	Y	Y	Y	100%
Kondylakis et al. 2020 [33]	Y	N	Y	Y	N	60%
Costantinescu et al. 2017 [26]	Y	N	Y	Y	Y	80%

**Table 3 healthcare-11-03116-t003:** Figures involved in gamification.

Author(s) and Year	Nurses (Oncology or Researcher)	Psychological Specialists	Other Healthcare Professionals	Researchers	Software Developers	Oncologists
Carcioppolo et al., 2022 [24]					✓	
Horsham et al., 2021 [25]				✓	✓	
Kim, Kim, Hwang et al., 2018 [27]		✓				
Loerzel, Clochesy and Geddie, 2018 [31]	✓					
Maganty et al., 2018 [28]			✓		✓	
Reichlin et al., 2011 [32]	✓			✓		
Loerzel, Clochesy and Geddie, 2020 [23]	✓			✓		✓
Cosma et al., 2016 [30]	✓				✓	
Thomas et al., 2019 [29]	✓	✓				✓
Constantinescu et al., 2017 [26]	✓		✓			
Kim; Kim; Shin et al., 2018 [21]		✓	✓			
Kondylakis et al., 2020 [33]	✓		✓		✓	
Khalil et al., 2016 [22]	✓			✓		

**Table 4 healthcare-11-03116-t004:** Game elements.

Author(s) and Year	Time Pressure and Competition	Shooting	Cards Match	Realistic Dialog and Response Options	Levels, Rewards, and Story Progression	Avatars
Carcioppolo et al., 2022 [24]			✓		✓	✓
Horsham et al., 2021 [25]	✓	✓			✓	✓
Kim, Kim, Hwang et al., 2018 [27]	✓	✓			✓	
Loerzel, Clochesy and Geddie, 2018 [31]					✓	✓
Maganty et al., 2018 [28]			✓			✓
Reichlin et al., 2011 [32]			✓			
Loerzel, Clochesy and Geddie, 2020 [23]					✓	✓
Cosma et al., 2016 [30]				✓	✓	✓
Thomas et al., 2019 [29]				✓		✓
Constantinescu et al., 2017 [26]				✓		
Kim; Kim; Shin et al., 2018 [21]						✓
Kondylakis et al., 2020 [33]		✓				✓
Khalil et al., 2016 [22]		✓				

## Data Availability

The data that support the findings of this study are available from the corresponding author, G.V., villa.giulia@hsr.it.

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
