# Peer review of "Gamification as an Educational Approach for Oncological Patients: A Systematic Scoping Review"

_healthcare, 2023, doi:10.3390/healthcare11243116_

Round 1

Reviewer 1 Report

Comments and Suggestions for Authors

First and foremost, I would like to thank you for the opportunity to review this important research paper. This study highlights the transformative potential of gamification in oncology by highlighting its effectiveness in engaging patients and improving biopsychosocial outcomes, providing a refreshing perspective to traditional methods of patient education. Given its importance, I would like to offer some constructive comments to the authors to further improve and increase the quality of this study. 

Major comments:

Materials and Methods

The Materials and Methods section is generally thorough and provides a clear roadmap for the process used. However, some suggestions can be made to further improve the clarity, coherence, and depth of this section:

1. Introductory Statement: begin with a more explicit introductory statement to set the context. For example, "Systematic scoping reviews are designed to provide a comprehensive overview of the existing literature on a particular topic. In this review, we aim to provide an overview of research on gamification in oncology education."

2. Details on JBI and PRISMA ScR: While you mention using the Joanna Briggs Institute (JBI) guidelines and the PRISMA ScR checklist, you should also provide a brief description of their relevance or importance in the context of systematic reviews.

3. Clarity of the research question: refine the research question to be more specific. For example, “To what extent is gamification effective in improving educational outcomes for cancer patients and survivors in diverse settings such as hospitals, homes, and territories?”

4. Elaborate on the PCC framework: mention the Participant Problem/Concept/Context (PCC) framework and provide a brief description or rationale for choosing this framework.

5. Exclusion Criteria Details: Provide a rationale for each exclusion criterion for clarity. For example, explain why you excluded studies of games with a purely gamified or entertainment function.

6. Search strategy: explain the rationale for the three-step search strategy. Why was each step taken and what was its significance?

7. Document selection: if applicable, indicate any challenges or ambiguities encountered in the document selection process.

8. Methodological quality: explain why the “Dixon-Woods Prompts for assessing quality in primary research” were chosen. Were any adjustments made or special considerations made in their application?

9. Data extraction: explain any challenges or decisions made during data extraction. If there were specific criteria to determine which data should be extracted, explain them.

Discussion

Add another subsection to the Discussion:

4.6. areas for improvement and recommendations

While this review has comprehensively examined gamification as an educational approach in oncology, it has also identified several areas for improvement and further research:

1. Diversity of study populations: Most of the articles reviewed focus predominantly on Western cultures, particularly the United States. This limitation limits the transferability and relevance of the findings to global contexts. Future research needs to examine the impact of gamification in diverse cultural and geographic settings to ensure broad applicability.

2. Standardization of measurement tools: The use of different instruments and scales to assess the effectiveness of serious games limits the ability to compare the results of different studies. A standardized set of outcome measures for evaluating gamification interventions would allow researchers to consistently compare and aggregate results from different studies.

3. Study Design and Sample Size: Many studies have limited and specific sample sizes, which compromises the generalizability of findings. Research with larger and more diverse samples is essential. Randomized controlled trials can provide more robust evidence of the effectiveness of gamification interventions.

4. Tailored Gamification Interventions: Although gamification holds promise in oncology education, it's important to ensure that game elements are tailored to specific cancer types, stages, and patient demographics. One-size-fits-all solutions may not meet the unique educational needs of different patient populations.

5. Consideration for the geriatric population: Although gamified interventions may be beneficial for older adults, their design should take into account potential challenges related to technology use, cognitive impairment, or sensory deficits. Research should further investigate the feasibility and acceptability of gamified interventions in older oncology patients.

6. Consider Potential Negative Effects: The potential negative effects of prolonged VR use, such as dizziness and balance disorders, need to be thoroughly investigated. Game developers should incorporate safety features to limit continuous use and provide regular breaks.

7. Inclusion of patient feedback: Involving patients in the design and evaluation of games can ensure that tools meet their educational needs, are user-friendly, and are culturally relevant.

8. Extending beyond disease education: While many games focus on disease education, there are opportunities to explore other aspects of patient care, such as mental health support, rehabilitation exercises, and post-treatment care.

In summary, while gamification is a promising avenue for improving oncology education, researchers and practitioners need to address the gaps identified. Tailored, culturally sensitive, and evidence-based gamification interventions can contribute significantly to improving patient education, engagement, and outcomes in oncology care.

Conclusions

I am suggesting some adjustments on this section.

5.1. Implications and recommendations for future research

This review has highlighted the potential of gamification as a complementary tool for patient education in oncology. However, there are still several opportunities to explore and improve:

More in-depth research: although initial results of gamification in oncology education are promising, more in-depth research is needed. Prospective cohort studies and randomized controlled trials should be conducted to determine the long-term benefits and potential harms of such interventions.

Diversity of research populations: As emphasized earlier, the cultural and geographic limitations in the current literature underscore the need for more global research. Understanding cultural nuances may enable the development of tailored serious games that target diverse patient populations.

Multimodal approach: while gamification offers significant benefits, it should not be considered a stand-alone approach. Integrating game-based interventions with traditional educational tools can provide a comprehensive and more effective patient education strategy.

Training for healthcare professionals: As gamification tools become more prevalent, healthcare professionals need training on how to effectively integrate and use these tools in their practice. This includes understanding game mechanics, monitoring patient progress, and adjusting educational strategies based on game feedback.

Stakeholder engagement: Engaging stakeholders, including patients, caregivers, and medical professionals, during the game development and evaluation process can ensure that games are relevant, user-friendly, and effective.

Economic evaluation: cost-effectiveness analyzes can provide insights into the economic benefits of integrating gamification interventions. This can help healthcare organizations make informed decisions about implementing such tools.

Longer follow-up periods: Longer-term studies can provide insight into whether the benefits of gamified interventions, such as reduced anxiety and improved symptom management, persist over time.

5.2. Final thought

Gamification in oncology education is a promising complementary tool for improving patient outcomes. Although preliminary results are encouraging, a comprehensive and multifaceted research approach is needed to truly understand the breadth and depth of its impact. Healthcare professionals remain at the center of the educational process, but with the right tools and training, gamified interventions can enhance their efforts and lead to better patient empowerment and improved care outcomes.

I would also add a Limitations section. 

Minor comments: 

Please check for punctuation mistakes (e.g., line 382). 

Author Response

Dear Reviewer

Thank you for your time and your suggestion. Attached the answer to your notes.

Regards

The authors

Reviewer 2 Report

Comments and Suggestions for Authors

The submitted manuscript explores a rather important and sensitive topic from a new perspective, approaching it from the standpoint of gamification. However, the manuscript needs significant revisions to be acceptable. A more detailed description is necessary for both the inclusion criteria and the exclusion criteria. Table 1 (which is missing the table format) does not provide sufficient information. The Quality Assessment is extremely deficient, and missing Risk of Bias tables. Additionally, studies with different methodologies need to be assessed in varying ways. I recommend reconsidering the quality assessment based on the following literature:

Modesti, P.A.; Reboldi, G.; Cappuccio, F.P.; Agyemang, C.; Remuzzi, G.; Rapi, S.; Perruolo, E.; Parati, G.; ESH Working Group on CV Risk in Low Resource Settings. Panethnic Differences in Blood Pressure in Europe: A Systematic Review and Meta-Analysis. PLoS ONE 2016, 11, e0147601.

Quality Assessment for the Systematic Review of Qualitative Evidence. https://www.ncbi.nlm.nih.gov/books/NBK262835/

A Guide to Evidence Synthesis: 9. Risk of Bias Assessment. https://guides.library.cornell.edu/evidence-synthesis/bias

RoB 2 for Cluster-Randomized Trials. https://www.riskofbias.info/welcome/rob-2-0-tool/rob-2-for-cluster-randomized-trials.

ROBINS-I Tool (Risk of Bias in Non-randomized Studies—Of Interventions). https://www.riskofbias.info/welcome/home

Revised Cochrane Risk of Bias Tool for Randomized Trials (RoB 2.0). https://www.riskofbias.info/welcome/rob-2-0-tool/archive-rob-2-0-2016.

The PRISMA Flow chart is not informative. Some paragraphs are in inappropriate locations (e.g. 3.3. and 3.3.1.) and it may not be necessary to dedicate an entire subsection to them (e.g. 3.3.).

In my opinion, the authors are using a very limited number of sources compared to what is typically expected for a systematic review.

Author Response

(The authors gave the same response as above.)

Reviewer 3 Report

Comments and Suggestions for Authors

Dear Author

Thank you for the work you present.

It has been very enjoyable to read. 

I would like to make a couple of points, in the spirit of improving the quality of your work.

- Were the selected papers read in a peer review process? Could you indicate the other author who participated? If not, could you indicate why?

Kinds Regards

Author Response

(The authors gave the same response as above.)

Reviewer 4 Report

Comments and Suggestions for Authors

Authors proposed an interesting review on the use of gamification in oncology for education purposes. The topic is in line with the journal's aims and the text is easy to read and useful for the research groups focused on VR and gamification in different fields. The methodology for the conduction of the scoping review should be better described. For example, authors should report the period and not only the starting of the searching, as well as to include in the inclusion criteria the language and the publication period. In addition, authors must indicate the keywords used during the searching.

For improving the attractiveness of the review I also suggest to include a paragraph on the future challanges and open questions of the topic.

Author Response

(The authors gave the same response as above.)

Round 2

Reviewer 1 Report

Comments and Suggestions for Authors

I would like to express my sincere gratitude to you for taking the time to consider my comments as a reviewer for your manuscript. Your receptiveness to feedback is commendable, and it is clear that you are committed to improving the quality of your work. Your willingness to engage in constructive dialogue not only promotes scholarly discourse overall, but also reflects your commitment to making a solid and valuable contribution to the field. Thank you for your appreciation of the peer review process and for your diligence in addressing the points raised.

Reviewer 4 Report

Comments and Suggestions for Authors

Authors properly replied to all my previous comments.